# Population-based socio-demographic household assessment of livelihoods and health among communities in Migori County, Kenya over multiple timepoints (2021, 2024, 2027): A study protocol

Joseph R. Starnes[1,2]*, Jane Wamae[2], Vincent Okoth[2], Daniele J. Ressler[2], Vincent Were[3], Lawrence P. O. Were[4,5], Troy D. Moon[6], Richard Wamai[7]

1 Department of Pediatrics, Vanderbilt University Medical Center, Nashville, TN, United States of America, 2 Lwala Community Alliance, Rongo, Migori County, Kenya, 3 Center for Geographic Medicine Research, Kenya Medical Research Institute, Nairobi, Kenya, 4 Department of Health Sciences, Sargent College of Health and Rehabilitation Sciences, Boston University, Boston, MA, United States of America, 5 Department of Global Health, Boston University School of Public Health, Boston, MA, United States of America, 6 Vanderbilt Institute for Global Health, Vanderbilt University Medical Center, Nashville, TN, United States of America, 7 Department of Cultures, Societies, and Global Studies, Northeastern University, Boston, MA, United States of America

* Joseph.Starnes@vumc.org

## Abstract

Migori County is located in western Kenya bordering Lake Victoria and has traditionally performed poorly on important health metrics, including child mortality and HIV prevalence. The Lwala Community Alliance is a non-governmental organization that serves to promote the health and well-being of communities in Migori County through an innovative model utilizing community health workers, community committees, and high-quality facility-based care. This has led to improved outcomes in areas served, including improvements in childhood mortality. As the Lwala Community Alliance expands to new programming areas, it has partnered with multiple academic institutions to rigorously evaluate outcomes. We describe a repeated cross-sectional survey study to evaluate key health metrics in both areas served by the Lwala Community Alliance and comparison areas. This will allow for longitudinal evaluation of changes in metrics over time. Surveys will be administered by trained enumerators on a tablet-based platform to maintain high data quality.

## Introduction

Migori County is located in western Kenya bordering Lake Victoria (Fig 1). One of the 47 counties in Kenya, Migori has historically underperformed on many important health metrics. For example, the under-five mortality rate in 2014 was 82 per 1,000 live births compared with 52 per 1,000 live births for Kenya as a whole [1]. HIV prevalence as of 2018 was 13% in Migori County compared to 4.9% nationally [2]. Per the 2014 Kenya Demographic Health Survey

**Funding:** The authors received no specific funding for this work. The research is funded out of the operating budget of the Lwala Community Alliance.

**Competing interests:** The authors have declared that no competing interests exist.

(DHS), only 57.2% of children in Migori County were fully vaccinated compared with 74.9% for Kenya as a whole [1]. Importantly, most data on these health indices come from national surveys such as the DHS and are disaggregated only to the regional or county level. Hyperlocal data to inform programming efforts in smaller areas is frequently not available.

The Lwala Community Alliance (Lwala) is a non-governmental organization that serves to promote the health and well-being of communities in the Rongo sub-county of Migori County, Kenya (Fig 2). Founded in 2007, Lwala initially worked in the community of North Kamagambo ward of Rongo sub-county with services including the operation of a hospital and clinic with inpatient, outpatient, maternal, and HIV care, as well as an innovative Community Health Worker (CHW) program incorporating traditional birth attendants. The CHW program is distinguished by its consistent payment, supportive supervision, and proactive community case finding and case management. This community health worker structure is supported by community committees that plan health initiatives and advocate for child rights, reproductive rights, sanitation infrastructure, and reduced HIV stigma.

Lwala's efforts in North Kamagambo have led to significant successes in several key health metrics, including being on track to outpace Millennium and Sustainable Development Goal (SDG) targets for childhood mortality, attaining an under-five mortality rate of 29.5 per 1,000 live births as of 2017 [3] compared to the SDG target of under 25 deaths per 1,000 live births by 2030 [4]. This success has led to local health authorities inviting Lwala to expand its CHW model into nearby East and South Kamagambo wards, as well as to begin providing technical assistance toward improved health service delivery in the government-supported health facilities of these wards. Along with service expansion has come the effort to systematically and academically evaluate health metrics and changes in outcomes in the community. This led to multiple iterations of a community-wide household survey in Lwala's original catchment area and subsequently in the expansion areas [3,5,6]. We now aim to conduct repeated cross-

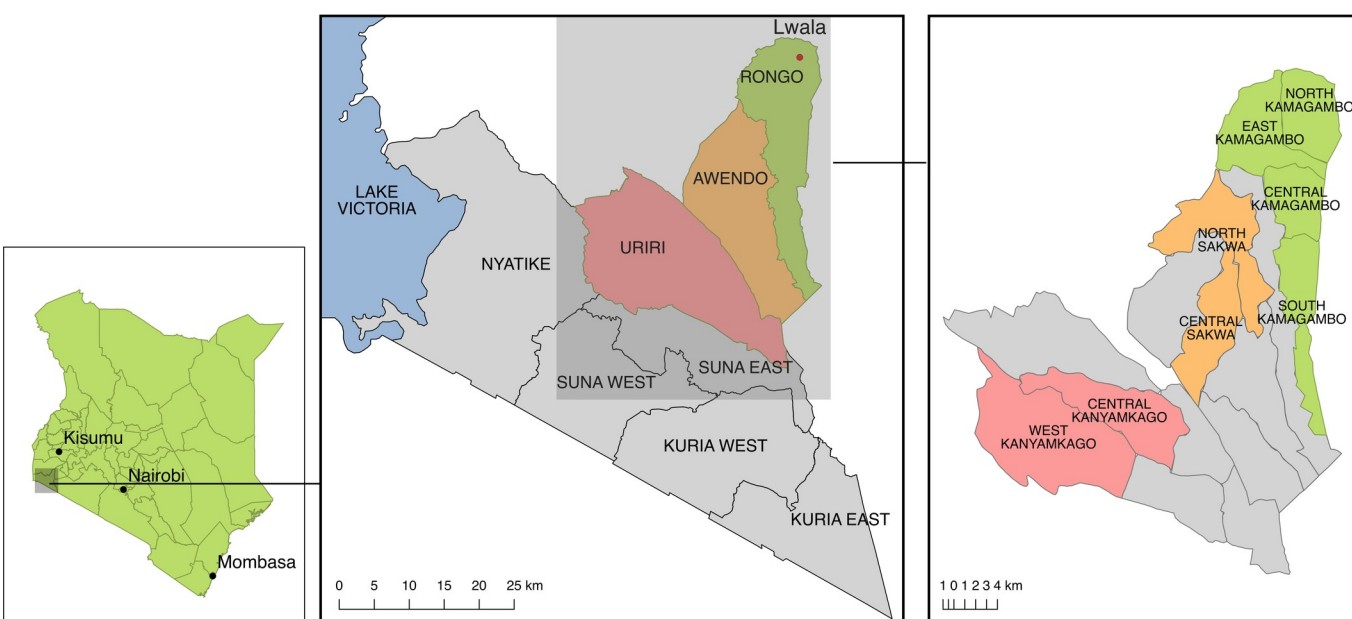

**Fig 1. Migori County, Kenya.** Lwala programming began in North Kamagambo in Rongo sub-county (green). By 2021, programming will include all of Rongo sub-county. The next expansion is planned for Awendo (orange). Two areas in Uriri, Central Kanyamkago and West Kanyamkago, serve as comparison wards (red).

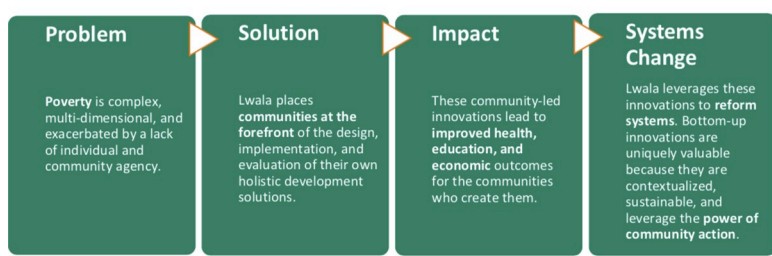

**Fig 2. Lwala Community Alliance theory of change.** Lwala utilizes a community-led model to improve health, education, and economic outcomes.

sectional surveys to evaluate the effect of Lwala's programming over time and characterize health metrics in the area.

Here we describe the Lwala Household Survey (LHS) with the following objectives: (1) to assess the health, socioeconomic, and education status of current and future communities receiving programming from Lwala; (2) to measure changes in these metrics over time in the presence of Lwala programming; (3) to compare changes over time in key metrics to determine the effect of Lwala presence and programming.

## Methods

### Study design

This is a repeated, cross-sectional survey allowing for longitudinal analyses of population and community-level metrics related to health, education, and socioeconomics. Households will be selected for surveying in areas currently receiving Lwala programming and in areas planned to receive Lwala programming in the future. Nearby geographic regions with no planned Lwala services will be surveyed and serve as comparison locations. Subsequent surveys will provide new cross-sectional data for a given geographic area by randomly selecting households for interview utilizing the same sampling strategy; however, it will not specifically target the same households for future surveying. Data collection will begin in 2021 and occur every three years until 2027. Previous surveys have been conducted in 2017 and 2019.

### Study population, setting, and timeline

Located in the Lake Victoria region of southwestern Kenya, Migori county had a population of 1.1 million people during the 2019 National Census [7]. Administratively, there are 10 sub-counties, each with multiple smaller electoral sub-divisions called wards [8]. With an average household size of 4.6 and a growth rate of 3.1%, it is highly densely populated (427 persons per square kilometer) [7,8]. About 90% of the population live in rural areas, largely in mud-walled houses with agriculture and fishing as the main livelihoods [8].

Lwala programming began in North Kamagambo, an area covering 46.4 square kilometers, in Rongo, which is one of the 10 sub-counties, in 2007 (Fig 1). Programs were subsequently expanded to other wards in Rongo, namely East Kamagambo in 2018 and South Kamagambo in 2019. Programming will be expanded to Central Kamagambo within Rongo in 2021 following the first survey administration. Lwala's area of programming is expanding approximately every two years, with the next round of expansions planned for Awendo sub-county. The first iteration of this survey in 2021 will be administered in current programming wards in Rongo sub-county (North Kamagambo, East Kamagambo, South Kamagambo) and the future

**Table 1. Survey timepoints across areas.**

| Sub-County | Ward | Intervention | 2017* | 2019* | 2021 | 2024 | 2027 |
|---|---|---|---|---|---|---|---|
| Rongo | North Kamagambo | 2007 | X | X | X | X | X |
| | East Kamagambo | 2018 | | X | X | X | X |
| | South Kamagambo | 2019 | | X | X | X | X |
| | Central Kamagambo | 2021 | | X | X | X | X |
| Awendo | North Sakwa | 2022 | | | X | X | X |
| | Central Sakwa | 2022 | | | X | X | X |
| Uriri | Central Kanyamkago | Comparison | | X | X | X | X |
| | West Kanyamkago | Comparison | | X | X | X | X |

*Surveys conducted as a part of previous works.

programming ward in Rongo sub-county (Central Kamagambo). It will also include two representative wards in Awendo sub-county (North Sakwa and Central Sakwa).

The initial survey will also include two comparison wards in Uriri sub-county (Central Kanyamkago and West Kanyamkago). Uriri sub-county is adjacent enough to be a comparable location but distant enough to minimize spillover effects. In addition, Uriri sub-county has a similar socio-economic and demographic context that is analogous to Rongo sub-County. Within Uriri sub-County, Central Kanyamkago was selected as a peri-urban ward to serve as a comparison with the peri-urban Central Kamagambo. Similarly, West Kanyamkago was selected as a rural ward to compare with more rural programming wards. While Uriri has government health facilities that are typical of Migori County, there is no similar organization to Lwala.

All subsequent surveys (2024 and 2027) will include the areas from the initial 2021 survey (Table 1). Any further expansion areas that are identified will also be included. Inclusion of new wards will be approved by the investigators with subsequent amendments made to Institutional Review Board protocols.

## Sample size

The study has a wide range of indicators of interest, including child mortality, skilled delivery rate, vaccination coverage, contraceptive prevalence, and antenatal care. These metrics vary in their community prevalence and would thus require different sample sizes. For example, under-five mortality is relatively rare at 82 per 1,000 live births while full vaccination is relatively common at 57.2% of children [1]. A community prevalence of 50%, yielding maximum variation and therefore maximum sample size, was used to adequately power all metrics. Within each area, the sample size was calculated to detect a 10% difference over time using a power of 80%, precision of 0.05, and design effect of 1.6. Design effect was calculated according to the equation:

$$DE = (1 + (m - 1)) * ICC$$

where DE is the design effect, m is the number of the household to be sampled per cluster, and ICC is the inter-cluster correlation. An ICC of 0.15 was used based on international standards [9]. This would require a sample of 621 households in each area. This estimate was inflated by 30% to give a total goal sample of 887 per area. For the first survey, which includes eight areas, the total sample size will be 7,096 households. Subsequent surveys will include the same number of households per area. Table 2 below shows the sample size for each ward.

**Table 2. Sample sizes for successive surveys.**

|  | Minimum | Maximum | Number of clusters | Households with children | Additional Households | Total per Cluster |
|---|---|---|---|---|---|---|
| North Kamagambo | 621 | 887 | 127 | 5 | 2 | 7 |
| East Kamagambo | 621 | 887 | 127 | 5 | 2 | 7 |
| Central Kamagambo | 621 | 887 | 127 | 5 | 2 | 7 |
| South Kamagambo | 621 | 887 | 127 | 5 | 2 | 7 |
| North Sakwa | 621 | 887 | 127 | 5 | 2 | 7 |
| Central Sakwa | 621 | 887 | 127 | 5 | 2 | 7 |
| Central Kanyamkago | 621 | 887 | 127 | 5 | 2 | 7 |
| West Kanyamkago | 621 | 887 | 127 | 5 | 2 | 7 |

## Sampling strategy

Households will be selected using a hybrid sampling technique to obtain as random of a sample as is feasible. Because a truly random sample would be logistically infeasible due to expense and lack of a household-level sampling frame, a hybrid systematic and random sampling technique will be used. To accomplish this, a modified procedure based on the World Health Organization Expanded Programme of Immunization (EPI) method will be used [10,11]. First, each area will be split into 127 grid squares using Geographic Information System (GIS) technology. The center point of each grid cell will then be generated using GIS. This is the starting location for the enumerators for each day's survey. GPS will be used to navigate to the precise starting location each morning.

After arrival at the center point, the spin-the-bottle technique will be used [10]. Each enumerator team will be supplied with a random direction by spinning a pen or bottle. On arrival to the center of the grid cell, they will travel in the given direction surveying houses along the line given by the direction.

As many of Lwala's programs and outcomes of interest focus around maternal and child health, households with children under five years of age will be oversampled. At least five of seven surveys administered in each grid square will be administered to households with children under five years. If the enumerator reaches the end of the grid cell before surveying both seven total households and five households with a child under five years, they will walk along the edge of the grid to the closest corner to find more households.

This approach minimizes the biases of the traditional spin-the-bottle sampling method [12] by using the center of an arbitrary square in place of the center of a town or gathering area.

## Survey instrument

The survey tool contains over 300 questions and is based on several different validated tools (Table 3). The full survey is available in the supplemental materials (S1 Appendix). The survey is designed to capture metrics across 13 public health modules in a reproducible manner. The estimated time to complete one interview is 45 minutes.

## Participant recruitment and enrollment procedures

Upon arrival to a selected household, the enumerator will first ask to speak to the head of the household. If the head of household is present, the enumerator will then ask if they have children under 5 years old living in the household. If no head of household is present, the enumerator will skip this house, going to the next household along the line selected. They will return to the household later if the head of household is returning home. We define a household as a

**Table 3. Survey modules, key metrics, and sample sizes.**

| Survey Section | Question Sources | Key Metrics |
|---|---|---|
| Personal Demographics | Kenya Demographic and Health Survey [1] | Age<br>Marital status |
| Household Information | Kenya Demographic and Health Survey [1] | Household census<br>Birth history (child mortality)<br>Total fertility rate |
| Economics | Kenya Demographic and Health Survey [1]<br>Poverty Probability Index [13] | Poverty probability<br>Multidimensional poverty index<br>Income |
| Family Planning | Kenya Demographic and Health Survey [1]<br>Condom Use Self-Efficacy Scale [14] | Contraceptive prevalence rate<br>Unmet need for contraception |
| Child Health | Kenya Demographic and Health Survey [1] | Antenatal care visits<br>Careseeking for childhood illness |
| Nutrition | WHO Infant and Young Child Feeding [15]<br>Household Hunger Scale [16] | Ever breastfed<br>Exclusive breastfeeding<br>Minimum acceptable diet<br>Household hunger |
| Vaccinations | Kenya Demographic and Health Survey [1] | Complete vaccination rate |
| HIV | Kenya Demographic and Health Survey [1]<br>van Rie Stigma Scale [17–19] | HIV testing rate<br>HIV stigma |
| Water and Sanitation | Kenya Demographic and Health Survey [1] | Drinking water source<br>Previous sanitation training |
| Education | Kenya Demographic and Health Survey [1] | School attendance<br>Educational attainment |
| Interpersonal Violence | Kenya Demographic and Health Survey [1]<br>Abuse Assessment Screen [20]<br>Partner Violence Screen [21] | Interpersonal violence prevalence |
| Mental Health | Patient Health Questionnaire (PHQ-8) [22,23] | Depressive symptom prevalence |
| Programming | Lwala monitoring and evaluation tools | Service access<br>Service satisfaction |
| Observational | Kenya Demographic and Health Survey [1] | Mosquito net use<br>Handwashing facility<br>Latrine type |
| COVID-19 | WHO COVID Survey Tool and Guidance [24]<br>van Rie Stigma Scale (adapted) [17–19] | Personal COVID-19 experience<br>Prevention behaviors<br>Testing and vaccination perceptions<br>COVID-19 stigma |

group of people that eat under the same roof that have lived in the same dwelling for the past year. This excludes temporary visitors.

Heads of households that are 18 years of age or older will be surveyed. Female heads of household are preferred as female family planning, interpersonal violence, and child health and nutrition are key survey areas. Male heads of household will only be surveyed if female heads are unavailable. The only other inclusion criteria will be living in a household in one of the surveyed communities. Households with children under five years of age will be over-sampled. Participants will receive 50 KES (about $0.50) in airtime for their participation.

## Ethical considerations

The protocol and study design for our household survey was approved by the Ethics and Scientific Review Committee at AMREF Health Africa on March 29, 2021 (AMREF-ESRC P452/2018) and the Institutional Review Board at Northeastern University on September 21, 2020 (IRB #: 20-09-18). A research permit was obtained through the National Commission for

Science, Technology and Innovation in Kenya on February 11, 2021 (NACOSTI/P/21/8776). All study personnel will undergo ethical research training.

## Safety and privacy

Prior to data collection, enumerators will obtain informed consent from each respondent in the form of a signature or thumbprint after reading a standardized script informing the respondent of the survey's purpose and confidentiality policy. Respondents who cannot read or write will be requested to invite a witness to participate in the consenting process. Potential participants are then encouraged to ask questions about the household survey before signing the consent form. Minors (below age 18) will not be surveyed, and consent will only be obtained from adults. For sensitive survey questions, specifically questions regarding mental health and interpersonal violence, an additional female enumerator will be available if female respondents prefer. Respondents will also be provided a contact number for a mental health counselor if concerns are identified.

Data from survey responses will primarily exist as digital copies. If paper surveys are administered due to technology failure, they will be entered into the electronic tool as soon as possible. Paper surveys will be kept in a locked, secure area. Data will be temporarily stored on individual tablets that are password-protected. Upon completion of the survey, data will be uploaded to a secure, privacy-protected online server. Enumerators are required to sign a form declaring that they will keep information obtained confidential and will undergo privacy training prior to survey administration. Interviews will be conducted in as private a location as available in the setting to avoid breaching respondent privacy during the survey itself.

The survey in 2021 will be conducted in the ongoing context of the COVID-19 pandemic. Standard operating procedures have been established to maximally diminish the risk of transmission and to protect both household participants and enumerators. Enumerators will be trained on transmission and prevention of COVID-19. All enumerators will be tested at the beginning of training, prior to survey implementation, and every two weeks during survey administration using a rapid diagnostic test (RDT). Enumerators will be screened for COVID-19-related symptoms each day [25], and any enumerator with symptoms will be referred for testing. If an enumerator tests positive, the Ministry of Health will be notified according to national guidelines [26]. Data collection teams and participants will be provided with sanitation materials and face masks. At all times, social distancing will be maintained between enumerators and participants. In addition, because most people spend their day outdoors and the survey takes place during daylight, interviews will be conducted primarily in outdoor settings to minimize risk of exposure. Each selected potential participant will be asked a series of COVID-19 exposure and symptom questions before the survey can begin. If there is concern for COVID-19 infection, this respondent will not be surveyed. At the end of the survey period enumerators will also be tested for COVID-19 using an RDT to assure no infection occurred during the survey. Lwala will conduct follow-up response per Ministry of Health guidelines, including notifying potentially exposed respondents.

## Enumerator selection, training, and team composition

Surveys will be administered in the household by trained enumerators who are not regular Lwala staff. All enumerators will be hired from the community. Enumerators will be selected from a pool of college graduates or equivalent experience, with preference given to those with experience in survey administration, Dholuo fluency, and high performance in training.

Prior to survey implementation, the enumerators will participate in a five-day training intended to familiarize them with the survey questions and tablet platform, the methodology

for household and respondent selection, and recommendations for dealing with potential challenges in the field. Training will focus on the responsible conduct of research, an understanding of the intent of the survey questions, the appropriate translation options in Dholuo, the appropriate skipping of questions according to survey logic, and the procedure for marking responses based on the question type. Enumerators that show signs that indicate the inability to interact appropriately with interviewees or generally do not perform well will be dismissed before field data collection begins.

Data collection teams will consist of a team leader and two enumerators. The team leader will assist in household identification and survey consent while enumerators are completing surveys with eligible households. An overall survey supervisor will also be present to assist with problems that arise and conduct spot checks by observing surveys.

### Data collection, management, and quality assurance

Enumerators will enter data on tablet-based questionnaires created using Research Electronic Data Capture (REDCap) [27,28]. REDCap is a secure, cloud-based software platform designed to support data capture for research studies. A new form will be created for each respondent, and forms will be submitted immediately upon completion of the interview. Paper surveys will be available but will only be used in the event of technology failure. All enumerators will be accompanied by a team leader to ensure accuracy. The survey is in English and will be translated into Dholuo, the most commonly spoken language in this population. Enumerators will use the translation version preferred by the respondent. The intent of each question will be established with enumerators during training prior to administration.

Risk of information sharing will be minimized through a password-protected tablet and mobile platform account with restricted device access. The data will be stored offline on the tablet application until synced to an online, privacy-protected server. All data will be uploaded to the online server for initial analysis through REDCap. Data quality checks will be conducted daily (Fig 3). Feedback from any data discrepancies will be shared with enumerators to maintain high-quality data entry. All variables will be checked line-by-line for any outliers. Surveys will be checked for internal validity, including checking for consistent answers regarding sex and household population. Surveys will also be checked for completeness and any missing data. Discrepancies in data will be resolved by the Data Management Team in conjunction with the enumerators involved. Any changes will be made through REDCap data management tools.

Raw data will be stored centrally in the REDCap online platform. At the conclusion of each survey round, raw data will be exported to Stata for data cleaning and processing. Data cleaning will be completed by the Data Management Team, and detailed records of all changes will be kept. A final, clean dataset will be kept in a central, password-protected location. All analyses will be conducted using this cleaned dataset.

### Data analysis

After administration, data will be exported to the latest version of Stata (StataCorp LP, College Station, TX) for further analysis. Initial analyses will focus on descriptive statistics of health, socioeconomic, and education metrics in the sample population across areas. Further analyses will characterize these metrics in terms of demographic variables and compare these metrics to county and national averages using appropriate statistical tests, including chi-squared tests, t-tests, ANOVA, and non-parametric tests. These analyses will be conducted following each survey in 2021, 2024, and 2027.

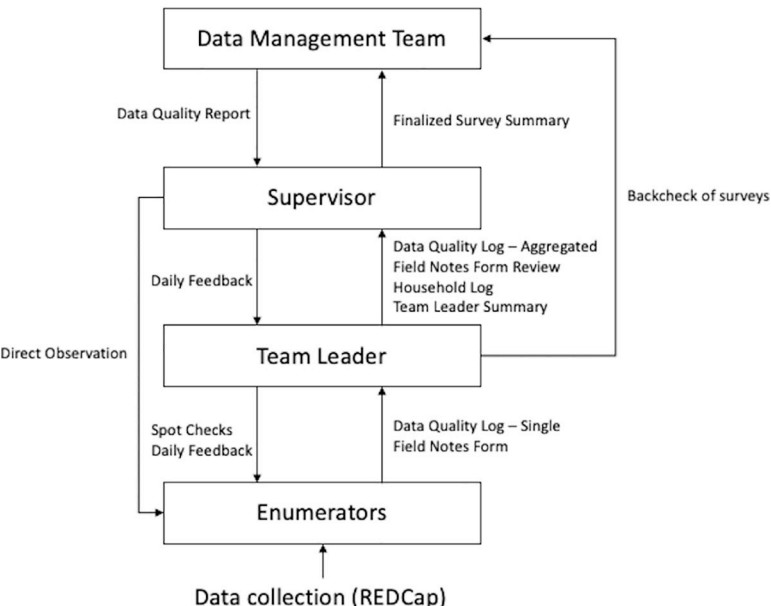

**Fig 3. Data quality plan.** Daily, bidirectional data quality checks will be conducted to ensure high-quality data collection. Enumerators will log any quality concerns and any issues during the survey day, which are reported to Team Leaders. Team Leaders review these forms and report to the Supervisor. The Supervisor then creates an aggregate summary that is distributed to the Data Management Team at the end of each survey day. The Data Management Team reviews the data present in REDCap each night for both internal validity and accuracy compared to field reports. This then forms the basis of a quality report that is reviewed with Supervisors and Team Leaders prior to the next survey day.

Analysis of outcome metrics will be conducted using multivariable linear regression, logistic regression, and generalized estimating equations. Specific primary outcomes of interest include under-five mortality, immunization rate, skilled delivery rate, contraceptive prevalence rate, and antenatal care visits. Longitudinal analyses will be performed at the end of the study in 2027 once multiple timepoints are available. Interim longitudinal analyses will also be performed following each survey administration. Trends in outcomes and potential effects of programs will be assessed. This will involve interrupted time series techniques with a segmented regression to asses intervention effects over repeated observations.

## Data availability

The data collected in this research project will be made available after finalization of the study together with corresponding statistical programming code upon request from the Lwala Community Alliance. All data shared will be anonymized.

## Discussion

### Limitations

Intrinsic to survey research is the use of self-reported data. Some of the survey questions employ direct observation, but the large majority of questions require the participant to recall information and offer sensitive opinions about themselves. We have minimized this limitation to the extent possible by using validated questions that include prompts and training enumerators in making participants comfortable. The cross-sectional nature of the survey makes determining causal relationships difficult. The use of repeated cross-sectional surveys and

sophisticated statistical methods will allow some commentary on causation, but the study will not have the same power as a randomized trial. However, randomization is not feasible given the unpredictable nature of expansion wards, which are chosen based on policy, organizational, and financial factors. Further, many of the quasi-experimental methods of program evaluation rely on the parallel trends assumption to create the counter-factual. While this study has a comparison group in the areas in Uriri, there is a single pre-implementation data point available for most intervention areas. This makes evaluation of the parallel trends assumption difficult. Finally, the repeated cross-sectional design loses power by not collecting longitudinal data for the same households over time. However, it was not logistically feasible to visit the same households at each timepoint. Phone numbers will be collected, and further studies will follow smaller sets of outcomes over time within households.

## Results dissemination

Data collected in the LHS will be disseminated for use by Lwala internal programming, to inform regional Migori county and national health policy makers, and will be shared widely through presentation at conferences and in the peer-reviewed literature. The reporting will be compliant with the Guidelines for Accurate and Transparent Health Estimates Reporting: the GATHER statement [29].

## Conclusion

We describe the proposed methods of the Lwala Household Survey to systematically evaluate public health metrics over time in Migori County, Kenya. This work will be carried out between 2021 and 2027 to inform ongoing programming efforts of the Lwala Community Alliance. Additionally, these results will be useful to regional and national programs and may also be applicable in similar settings outside Kenya. While smaller in size of population covered, the LHS will add to methodological designs and empirical field work of other household surveys, such as the national DHS and regional health and demographic surveillance networks [30,31].

## Supporting information

**S1 Appendix. Survey.** Complete survey as it will be administered, although survey has been digitized into a tablet-based program.
(DOCX)

## Author Contributions

**Conceptualization:** Joseph R. Starnes, Jane Wamae, Vincent Okoth, Daniele J. Ressler, Vincent Were, Lawrence P. O. Were, Troy D. Moon, Richard Wamai.

**Data curation:** Joseph R. Starnes, Jane Wamae, Daniele J. Ressler, Vincent Were, Lawrence P. O. Were, Troy D. Moon, Richard Wamai.

**Formal analysis:** Joseph R. Starnes, Jane Wamae, Vincent Were, Lawrence P. O. Were.

**Funding acquisition:** Daniele J. Ressler.

**Investigation:** Joseph R. Starnes, Vincent Okoth, Daniele J. Ressler, Vincent Were, Lawrence P. O. Were, Troy D. Moon, Richard Wamai.

**Methodology:** Joseph R. Starnes, Jane Wamae, Vincent Okoth, Daniele J. Ressler, Vincent Were, Lawrence P. O. Were, Troy D. Moon, Richard Wamai.

**Project administration:** Joseph R. Starnes, Jane Wamae, Vincent Okoth, Daniele J. Ressler, Vincent Were, Lawrence P. O. Were, Troy D. Moon, Richard Wamai.

**Resources:** Joseph R. Starnes, Daniele J. Ressler.

**Software:** Joseph R. Starnes, Vincent Okoth, Vincent Were.

**Supervision:** Joseph R. Starnes, Vincent Okoth, Daniele J. Ressler, Vincent Were, Lawrence P. O. Were, Troy D. Moon, Richard Wamai.

**Validation:** Joseph R. Starnes, Jane Wamae, Vincent Okoth.

**Writing – original draft:** Joseph R. Starnes.

**Writing – review & editing:** Joseph R. Starnes, Jane Wamae, Vincent Okoth, Daniele J. Ressler, Vincent Were, Lawrence P. O. Were, Troy D. Moon, Richard Wamai.

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
