## [Decision Letter · Decision Letter 0]

1 Jul 2021

PONE-D-21-14196

Population-based socio-demographic household assessment of livelihoods and health among communities in Migori County, Kenya over multiple timepoints (2021, 2024, 2027): A study protocol

PLOS ONE

Dear Dr. Starnes,

Thank you for submitting your manuscript to PLOS ONE. After careful consideration, we feel that it has merit but does not fully meet PLOS ONE’s publication criteria as it currently stands. Therefore, we invite you to submit a revised version of the manuscript that addresses the points raised during the review process.

We look forward to receiving your revised manuscript.

Kind regards,

Bidhubhusan Mahapatra, Ph.D.

Academic Editor

PLOS ONE

Journal Requirements:

1. Please ensure that your manuscript meets PLOS ONE's style requirements, including those for file naming. The PLOS ONE style templates can be found athttps://journals.plos.org/plosone/s/file?id=wjVg/PLOSOne_formatting_sample_main_body.pdf and https://journals.plos.org/plosone/s/file?id=ba62/PLOSOne_formatting_sample_title_authors_affiliations.pdf

Additional Editor Comments (if provided):

This is an important study protocol. A couple of minor suggestions that author should include in the revised version: (i) provide some details on data quality assurance; how exactly it is implemented at various stages starting from tool development, recruitment of investigators to data processing? (ii) Provide a detailed data management plan to illustrate how data flows from field and how it is made ready for analysis.

Reviewers' comments:

Reviewer's Responses to Questions

**Comments to the Author**

1. Does the manuscript provide a valid rationale for the proposed study, with clearly identified and justified research questions?

Reviewer #1: Yes

Reviewer #2: Yes

2. Is the protocol technically sound and planned in a manner that will lead to a meaningful outcome and allow testing the stated hypotheses?

Reviewer #1: Yes

Reviewer #2: Yes

3. Is the methodology feasible and described in sufficient detail to allow the work to be replicable?

Reviewer #1: Yes

Reviewer #2: Yes

4. Have the authors described where all data underlying the findings will be made available when the study is complete?

Reviewer #1: Yes

Reviewer #2: Yes

5. Is the manuscript presented in an intelligible fashion and written in standard English?

Reviewer #1: Yes

Reviewer #2: Yes

6. Review Comments to the Author

You may also provide optional suggestions and comments to authors that they might find helpful in planning their study.

Reviewer #1: The proposal for population-based socio-demographic household assessment of livelihoods and health

among communities in Migori County, Kenya over multiple timepoints is clear in its scope and is well-written. The language used is clear, however, it would have been useful if a conceptual diagram/ theory of change for the program was provided for the readers. While the proposal makes it a good case for the proposed study, I do have few questions:

1. The reason for carrying out a study over multiple points in time using repeated cross-sectional methods with control arm while not using a panel data even for few sections [such as child health] of the study was not quite clear. Have authors considered the possibilities of conducting a phone-based follow-up to capture the longitudinal impact of the program on selected outcomes?

2. While authors have mentioned the precautions and social distancing that will be taken during the survey, as a best practice would participants be provided with hand sanitizers/masks during the interview? This is a suggestion given that the survey will be conducted in a disadvantaged community that might not have enough resources and information to protect themselves from COVID-19. I do have similar concern for acquiring participant’s assent/consent by physically receiving participants’ signature or thumbprint. Have authors thought about any other ways of acquiring it [audio-taping]?

3. The survey aims to cover a broad range of indicators using variety of validated tools. I am little unsure about the estimated time [45 minutes] for over 300 questions, some of these being sensitive in nature.

4. It would be important to mention the state of the art for the comparison area. Are there programs similar to the program being implemented in the intervention arm being run by any other non-governmental organization that might impact the interpretation of the difference in the outcomes between intervention and the comparison arm?

5. For mental health outcomes measurement, authors have mentioned about the patient health questionnaire. It would be helpful to mention which version of this tool will be used [PHQ-9- specifically for capturing the depressive symptoms? Etc.].

6. Apart from measuring the community-level impact on the desired outcomes would like to know why does this evaluation lack any component that measures the quality of the service delivery, and the providers’ aspects given that the program specifically entails community health workers, community committees, and high quality facility-based care?

7. As authors mention that the program “serves to promote the health and well-being of communities in the Rongo sub-county”, wouldn’t a mixed-methods approach with focus group discussions combined with the quantitative surveys help understand the community level acceptance, challenges, and expectations from the program?

Reviewer #2: Overall, the proposed study is well-written and technically sound. The repeated cross-sectional design and the objective of capturing key indicators at household, child, and woman-level every three years until 2027 will make the findings from this study very useful for policy implications.

Please see few remarks below that can help improve data collection and quality.

a. A survey tool with 300 questions even with suitable checks and logics in place and covering various domains – interpersonal violence, mental health, IYCF- with so many questions based on recall will be difficult to complete in 45 minutes. The team should re-consider the questionnaire length to reduce the respondent fatigue thereby improving data quality. These are mothers with children <5 years and may not have time to answer ~300 questions.

b. Respondents should be informed and given a time range for example, time to complete interview as 45-60 minutes before starting the survey.

c. Will respondents receive any compensation for their time? They may ask how they benefit from participating in the study.

7. PLOS authors have the option to publish the peer review history of their article (what does this mean?). If published, this will include your full peer review and any attached files.

Reviewer #1: No

Reviewer #2: No

---

## [Author Response · Author response to Decision Letter 0]

29 Jul 2021

We very much appreciate this thoughtful feedback on our submission and are happy to make modifications accordingly. We have copied the reviewers’ and editor’s comments below and have given our responses, including any changes made to the manuscript, below each comment. We have also included both a version with tracked changes and a clean version of our manuscript with this submission. 

Editor Comments

This is an important study protocol. A couple of minor suggestions that author should include in the revised version: (i) provide some details on data quality assurance; how exactly it is implemented at various stages starting from tool development, recruitment of investigators to data processing? (ii) Provide a detailed data management plan to illustrate how data flows from field and how it is made ready for analysis.

We agree that readers need detailed information regarding data quality assurance and data management. We have added text to page 15 to help clarify our methods.

Reviewer Comments

Reviewer #1 

The proposal for population-based socio-demographic household assessment of livelihoods and health among communities in Migori County, Kenya over multiple timepoints is clear in its scope and is well-written. The language used is clear, however, it would have been useful if a conceptual diagram/ theory of change for the program was provided for the readers. 

We agree that a theory of change diagram will be helpful to readers. We have included this new figure and have updated figure numbering as appropriate. 

While the proposal makes it a good case for the proposed study, I do have few questions:

1. The reason for carrying out a study over multiple points in time using repeated cross-sectional methods with control arm while not using a panel data even for few sections [such as child health] of the study was not quite clear. Have authors considered the possibilities of conducting a phone-based follow-up to capture the longitudinal impact of the program on selected outcomes?

We agree with the reviewer that longitudinal data for selected outcomes would add additional power, but it was deemed logistically infeasible to visit the same households for each survey. Phone numbers will be collected from participants, which will allow for future longitudinal studies of more focused outcomes. We have added text to the Limitations (page 17, lines 327-331) to clarify this. Further, ongoing data collection by CHWs will allow for longitudinal data collection at the household level that is outside this study.

2. While authors have mentioned the precautions and social distancing that will be taken during the survey, as a best practice would participants be provided with hand sanitizers/masks during the interview? This is a suggestion given that the survey will be conducted in a disadvantaged community that might not have enough resources and information to protect themselves from COVID-19. I do have similar concern for acquiring participant’s assent/consent by physically receiving participants’ signature or thumbprint. Have authors thought about any other ways of acquiring it [audio-taping]?

We share the reviewer’s concerns about safety during COVID-19. Both enumerators and participants are provided with sanitization supplies during the survey. We have added text to page 12-13, lines 222-223 to clarify this. It is interesting to consider other ways of obtaining consent, but our IRB approval only provides for written consent. Given that the first round of surveying has been conducted while this manuscript was under review, we will revisit this possibility during the next timepoint if COVID-19 spread is still a concern. 

3. The survey aims to cover a broad range of indicators using variety of validated tools. I am little unsure about the estimated time [45 minutes] for over 300 questions, some of these being sensitive in nature.

We agree that the survey is quite long and covers a broad range of indicators. During our first administration over the past two months, survey times generally ranged from 35-45 minutes. We believe this quick administration was achieved largely due to extensive enumerator training and the use of an electronic tool with programmed skip logic.

4. It would be important to mention the state of the art for the comparison area. Are there programs similar to the program being implemented in the intervention arm being run by any other non-governmental organization that might impact the interpretation of the difference in the outcomes between intervention and the comparison arm?

We agree this is important to clarify for the reader. We have added text to page 6, lines 118-119. There is no organization similar to Lwala operating within the control areas. 

5. For mental health outcomes measurement, authors have mentioned about the patient health questionnaire. It would be helpful to mention which version of this tool will be used [PHQ-9- specifically for capturing the depressive symptoms? Etc.].

We have used the PHQ-8, which is the PHQ-9 without the final question regarding suicidal ideation. This question has been omitted due to community concern about the sensitive nature of this question. We have added this to Table 3 to clarify. 

6. Apart from measuring the community-level impact on the desired outcomes would like to know why does this evaluation lack any component that measures the quality of the service delivery, and the providers’ aspects given that the program specifically entails community health workers, community committees, and high quality facility-based care?

Although it is not immediately evident from Table 3, the survey does contain questions regarding services provided and satisfaction with services, both for home care provided by CHWs and facility-based care. These questions are within the Programming module. Lwala Community Alliance also has ongoing quality improvement projects at facilities in programming areas that more specifically capture quality metrics. 

7. As authors mention that the program “serves to promote the health and well-being of communities in the Rongo sub-county”, wouldn’t a mixed-methods approach with focus group discussions combined with the quantitative surveys help understand the community level acceptance, challenges, and expectations from the program?

We wholeheartedly agree that mixed-methods approaches allow deeper understanding of communities and community health. Our general approach has been to use quantitative data captured in the household survey to inform future mixed-methods projects in concern areas uncovered by the household survey. As data from the household survey is available, these projects will begin.

Reviewer #2

Overall, the proposed study is well-written and technically sound. The repeated cross-sectional design and the objective of capturing key indicators at household, child, and woman-level every three years until 2027 will make the findings from this study very useful for policy implications.

Please see few remarks below that can help improve data collection and quality.

a. A survey tool with 300 questions even with suitable checks and logics in place and covering various domains – interpersonal violence, mental health, IYCF- with so many questions based on recall will be difficult to complete in 45 minutes. The team should re-consider the questionnaire length to reduce the respondent fatigue thereby improving data quality. These are mothers with children <5 years and may not have time to answer ~300 questions.

As stated above, through extensive enumerator trainings and the use of an electronic data capture tool we were able to limit surveys to about 35-45 minutes. We hope to limit questionnaire length for future iterations by limiting questions related to COVID-19, which was a focus of this first survey timepoint.

b. Respondents should be informed and given a time range for example, time to complete interview as 45-60 minutes before starting the survey.

We agree that respondents should understand the length of the survey before beginning. This is included as a part of our consent process.

c. Will respondents receive any compensation for their time? They may ask how they benefit from participating in the study. 

Participants will receive 50 KES (~$0.50) in airtime for their participation. Because it is not uncommon for residents within the catchment area to work for $1 per day, we selected 50 KES to avoid coercing potential participants to participate for financial benefit. We have added text to page 11, line 185-186 to clarify this.

---

## [Editor Report · Decision Letter 1]

10 Aug 2021

Population-based socio-demographic household assessment of livelihoods and health among communities in Migori County, Kenya over multiple timepoints (2021, 2024, 2027): A study protocol

PONE-D-21-14196R1

Dear Dr. Starnes,

We’re pleased to inform you that your manuscript has been judged scientifically suitable for publication and will be formally accepted for publication once it meets all outstanding technical requirements.

Kind regards,

Bidhubhusan Mahapatra, Ph.D.

Academic Editor

PLOS ONE
---

## [Editor Report · Acceptance letter]

16 Aug 2021

PONE-D-21-14196R1 

Population-based socio-demographic household assessment of livelihoods and health among communities in Migori County, Kenya over multiple timepoints (2021, 2024, 2027): A study protocol 

Dear Dr. Starnes:

I'm pleased to inform you that your manuscript has been deemed suitable for publication in PLOS ONE. Congratulations! Your manuscript is now with our production department. 

Kind regards, 

on behalf of

Dr. Bidhubhusan Mahapatra 

Academic Editor

PLOS ONE